# Role of RET-Regulated GDNF-GFRα1 Endocytosis in Methamphetamine-Induced Neurotoxicity

**DOI:** 10.3390/ijms26199522

**Published:** 2025-09-29

**Authors:** Mengran Lv, Baoyu Shen, Zhenling Wu, Genmeng Yang, Yuanyuan Cao, Yuan Zhang, Junjie Shu, Wenjuan Dong, Zhenping Hou, Di Jing, Xinjie Zhang, Yuhan Hou, Jing Xu, Lihua Li, Shijun Hong

**Affiliations:** National Health Commission (NHC) Key Laboratory of Drug Addiction Medicine, School of Forensic Medicine, Kunming Medical University, Kunming 650500, China; lvmengran202103@163.com (M.L.); baoyu19950629@163.com (B.S.); m18388007920@163.com (Z.W.); yanggenmeng@kmmu.edu.cn (G.Y.); caoyuanyuan0099@163.com (Y.C.); 18487392405@163.com (Y.Z.); wangyi1914489429@163.com (J.S.); d1013754851@163.com (W.D.); 13783853476@163.com (Z.H.); jingdi2025@163.com (D.J.); jir0519@163.com (X.Z.); hyh1768066932@163.com (Y.H.); 20230059@kmmu.edu.cn (J.X.)

**Keywords:** methamphetamine, glia-derived neurotrophic factor, transmembrane receptor tyrosine kinase, neurotoxicity, endocytosis

## Abstract

Methamphetamine (METH) is a highly addictive synthetic psychostimulant that can induce severe neurotoxicity, leading to neurodegeneration similar to neurodegenerative diseases. The endocytosis of glial cell line-derived neurotrophic factor (GDNF) and its family receptor alpha 1 (GFRα1), regulated by transmembrane receptor tyrosine kinase (RET), has been shown to resist neurodegeneration. Specifically, the endocytosis of GDNF-GFRα1 mediated by RET is crucial in protecting neurons. Although many molecular mechanisms of METH induced neurotoxicity have been explored, the obstacles to the neuroprotective effect of GDNF in the context of METH induced neurotoxicity are still unclear. In this study, an increase in cell apoptosis and GDNF expression was observed in the hippocampus of METH abusers. METH also induces cell degeneration, cytotoxicity, and GDNF expression and release in hippocampal neuronal (HT-22) cells in a concentration-dependent manner (0.25, 0.5, 1, 2, and 4 mM) and time-dependent manner (3, 6, 12, 24, and 48 h). Meanwhile, after 24 h of exposure to METH (2mM), apoptosis, impaired endocytosis of GDNF-GFRα1, and decreased expression of RET were observed in HT-22 cells and organotypic hippocampal slices of mice. More notably, overexpression of RET weakened METH induced cell degeneration, apoptosis, and disruption of GDNF-GFRα1 endocytosis in HT-22 cells. This study suggests that RET is a key molecule for METH to disrupt GDNF-mediated neuroprotective signaling, and targeting RET-mediated endocytosis of GDNF-GFRα1 may be a potential therapeutic approach for METH induced neurotoxicity and neurodegeneration.

## 1. Introduction

Methamphetamine (METH), commonly referred to as crystal meth, is a highly addictive synthetic psychostimulant that presents a significant global public health challenge. Despite extensive efforts to address METH abuse, it remains a widespread issue requiring urgent intervention. Substantial evidence has shown that prolonged METH exposure can induce serious addiction, neurotoxicity, and neuroinflammation [1,2,3,4]. Furthermore, chronic METH neurotoxicity is implicated in neuropathological damage and the progression of neurodegenerative disorders [5,6,7] often leading to severe neuropsychiatric conditions. Moreover, METH-induced behavioral disorders are closely associated with hippocampal neural damage [8,9,10]. Although considerable research has explored the mechanisms underlying METH-induced neurotoxicity, the disruption of neuroprotective pathways remains poorly understood.

Glial cell-line-derived neurotrophic factor (GDNF) is a critical neurotrophic factor essential for neuronal development and survival, particularly within the neurotrophic factor family [11,12]. Under pathological conditions, GDNF is primarily produced by glial cells, including astrocytes and microglia [13,14]. Recent studies have shown that highly enriched astrocyte or microglial cultures can express GDNF, but lipopolysaccharide (LPS) stimulation failed to further increase its expression. However, the expression of GDNF in mixed cultured glial cells was significantly increased after LPS stimulation, suggesting that the interaction between microglia and astrocytes is a necessary condition for enhancing GDNF expression [15].

GFRα1, a key member of the GDNF receptor-α family (GFRα1), serves as the primary co-receptor for GDNF [16]. However, due to its lack of intracellular signaling domains, additional molecular components are required to propagate signals following GDNF binding. Studies have indicated that the GDNF-GFRα1 complex recruits transmembrane receptors, notably the receptor tyrosine kinase (RET). Subsequently, endocytosis mediated by clathrin promoted the internalization of RET receptors in vivo [17]. Although previous studies have reported increased GDNF expression in the rat substantia nigra following METH exposure [18,19], this up-regulation does not fully counteract METH-induced neurotoxicity. The mechanisms underlying this apparent failure remain unclear, necessitating further investigation. RET, a transmembrane receptor tyrosine kinase and key component of the GDNF signaling cascade, exhibits inherently weak affinity for GDNF. Its interaction with GFRα1 is required to form the GDNF-GFRα1-RET complex and recruit multiple downstream effectors for endocytosis into cells for intracellular signal transduction [17]. Intracellular signaling pathways that occur downstream of RET are responsible for essential cellular processes, including survival, differentiation, proliferation, migration, chemotaxis, neurite growth, and synaptic plasticity [17]. Recent studies have demonstrated that RET expression is significantly reduced in Parkinson’s disease models induced by 6-hydroxydopamine (6-OHDA) and 1-methyl-4-phenyl-1,2,3,6-tetramethylpyridine (MPTP) [20,21,22], indicating that neurotoxic insults can inhibit RET expression. Given the pivotal role of RET in mediating GDNF-GFRα1 endocytosis, we hypothesized that METH disrupts this process, thereby impairing the neuroprotective function of GDNF and exacerbating METH-induced neurotoxicity.

Endocytosis is a dynamic cellular process through which membrane-bound components and extracellular materials are internalized into cytoplasmic vesicles and trafficked to early and late endosomes. These vesicular structures either recycle their contents back to the cellular membrane or direct them to lysosomes for degradation [23]. Various approaches have been developed to investigate receptor internalization and surface protein dynamics in living cells. One widely used method involves membrane protein fractionation, in which total membrane proteins are isolated and analyzed via Western blot. By quantifying the proportion of membrane-bound receptors relative to total protein levels, this technique provides indirect evidence of receptor internalization [24]. A more direct approach is immunofluorescence-based receptor tracking, which enables visualization of receptor endocytosis at the single-cell level. In this method, a FLAG epitope is introduced into the target receptor via genetic knock-in, allowing selective labeling with an Alexa-488-conjugated M1 anti-FLAG antibody. Upon stimulation with the receptor’s ligand, internalization occurs over a defined time frame. A secondary Alexa-594-conjugated anti-species IgG is then applied to label residual surface receptors. This approach differentiates internalized (green-labeled) and membrane-retained (red-labeled) receptors, providing real-time insights into receptor trafficking dynamics [25]. Another established technique is biotin-based surface labeling, where sulfo-NHS-SS-biotin is introduced into the culture medium to selectively label cell surface proteins. Following ligand stimulation, biotinylated receptors undergo endocytosis. To distinguish internalized from surface-bound proteins, extracellular biotin is removed using 5 mM glutathione before cell lysis. Western blot analysis is then used to detect the internalized, biotinylated receptors, offering a quantitative assessment of endocytosis efficiency [25]. In this study, two complementary methods were selected: membrane protein fractionation and biotin-based receptor labeling.

## 2. Results

### 2.1. METH Induces Apoptosis and Up-Regulates GDNF Expression in the Human Hippocampus

The neurotoxic effects of METH and its impact on GDNF expression in the human hippocampus were evaluated by assessing apoptosis and GDNF levels in post-mortem hippocampal tissue. Compared to the control group, the number of apoptotic cells in the dentate gyrus (DG) was significantly higher in individuals with a history of METH abuse (Figure 1a, *p* < 0.001). In addition, METH exposure was associated with a marked increase in the pro-apoptotic protein Bax (Figure 1c, *p* < 0.05) and a significant reduction in the anti-apoptotic protein Bcl-2 (Figure 1b, *p* < 0.05). However, GDNF expression was significantly elevated in the hippocampus of METH abusers compared to the controls (Figure 1d,e, *p* < 0.05 or *p* < 0.001). These findings suggest that while METH induces apoptosis in human hippocampal neurons, it also triggers an increase in GDNF expression.

### 2.2. METH Induces Cell Degeneration, Cytotoxicity, and Increased GDNF Expression and Release in a Dose- and Time-Dependent Manner

To elucidate the mechanism underlying METH-induced neurotoxicity and the associated increase in GDNF expression, in vitro cell experiments were conducted using a co-culture system consisting of mouse microglial (BV2), astroglial (MA), and hippocampal neuronal (HT-22) cell lines at a ratio of 1:2:5. In the Transwell co-culture system, BV2 and MA cells were cultured in the upper chamber, while HT-22 cells were cultured in the lower chamber. Results showed that METH exposure resulted in a concentration-dependent (0.25, 0.5, 1, 2, and 4 mM) reduction in the long-to-short axis ratio of hippocampal neurons (0.5, 1, 2, and 4 mM) (Figure 2a,c, *p* < 0.001). Similarly, METH reduced cell viability in a concentration-dependent manner, with statistical significance reached at concentrations of 2 and 4 mM (Figure 2e, *p* < 0.001). Further analysis demonstrated that treatment with 2 mM METH reduced the long-to-short axis ratio (Figure 2b,d, *p* < 0.01 or *p* < 0.001) and decreased cell activity (Figure 2f, *p* < 0.001) in a time-dependent manner (3, 6, 12, 24, and 48 h), although there was only a significant difference at 48 h.

To assess GDNF expression, another Transwell co-culture system was employed, in which HT-22 and BV2 cells were cultured in the upper chamber and MA cells were seeded in the lower chamber. Results showed that METH exposure led to a significant concentration- and time-dependent increase in GDNF expression (Figure 3a,b, *p* < 0.01 or *p* < 0.001). In the direct contact co-culture system, both the expression and release of GDNF followed a similar concentration- and time-dependent pattern (Figure 3c–f, *p* < 0.001, *p* < 0.01, or *p* < 0.05). Based on comprehensive analyses of neuronal morphology, survival rate, and GDNF expression and release, METH exposure at 2 mM for 24 h was selected as the condition for subsequent experiments.

### 2.3. METH Induces HT-22 Cell APOPTOSIS, Disrupts GFRα1 Endocytosis, and Down-Regulates RET Expression

To further explore the neurotoxic effects of METH, apoptosis and GFRα1 endocytosis were assessed in HT-22 cells following 24 h exposure to 2 mM METH. It was observed that the pro-apoptotic protein Bax exhibited an increasing trend (Figure 2g). A significant up-regulation of Caspase-3 and cleaved Caspase-3 was observed (Figure 2g, *p* < 0.001), while anti-apoptotic Bcl-2 expression was markedly reduced (Figure 2g, *p* < 0.001), indicating enhanced apoptotic activity. In addition, METH exposure resulted in a significant accumulation of GFRα1 on the HT-22 cell membrane (Figure 4c, *p* < 0.001); however, total GFRα1 expression did not change significantly (Figure 4b, *p* > 0.05). Further experiments using biotin-labeled proteins demonstrated that GFRα1 endocytosis was reduced after METH exposure (Figure 4a,d, *p* < 0.05). This suggests that RET receptor expression was also significantly reduced following METH exposure compared to the control group (Figure 4e, *p* < 0.05). These findings indicate that METH-induced neurotoxicity is associated with increased GDNF expression and release from glial cells. However, GDNF does not exert a neuroprotective effect, which may be attributed to impaired GDNF-GFRα1 endocytosis, potentially exacerbated by reduced RET expression.

### 2.4. METH Induces Apoptosis, Up-Regulates GDNF Expression, and Down-Regulates RET Expression in Mouse Hippocampal Cells

To evaluate the neurotoxic effects of METH, a mouse model of acute toxicity was established, followed by the examination of hippocampal tissue. METH exposure for 24 h resulted in a significant increase in the expression of the pro-apoptotic protein Bax (Figure 5d, *p* < 0.05) and a marked elevation in the number of apoptotic cells (Figure 5e, *p* < 0.001). Conversely, the expression of the anti-apoptotic protein Bcl-2 was significantly reduced (Figure 5c, *p* < 0.05), indicating enhanced apoptotic activity. In addition, METH exposure led to a significant up-regulation of GDNF expression in the hippocampus (Figure 5a,b, *p* < 0.001 or *p* < 0.05), while RET expression was markedly suppressed (Figure 5f,g, *p* < 0.001 or *p* < 0.05).

### 2.5. RET Overexpression Attenuates METH-Induced HT-22 Cytotoxicity and GFRα1 Endocytosis Defects

To explore the effect of RET on GDNF-GFRα1 endocytosis during METH exposure, HT-22 cells overexpressing RET were constructed. The 80 MOI LV cell line was selected based on RET expression levels (Figure 6a, *p* < 0.01) for subsequent experiments. In the Transwell co-culture system, BV2 and MA cells were cultured in the upper chamber, while RET-overexpressing HT-22 cells were cultured in the lower chamber. Exposure to 2 mM METH for 24 h reduced the long-to-short axis ratio in hippocampal neurons (Figure 6b, *p* < 0.001) and decreased cell viability in the direct contact co-culture system (Figure 6c, *p* < 0.01), while compared to exposure to METH, RET overexpression increased the long-to-short axis ratio (Figure 6b, *p* < 0.01) and improved cell activity (Figure 6c, *p* < 0.01). METH exposure also significantly increased the expression of the pro-apoptotic proteins Bax, caspase-3, and cleaved caspase-3 (Figure 6h,j,k, *p* < 0.05 or *p* < 0.01) while decreasing Bcl-2 expression (Figure 6g, *p* < 0.05), resulting in an increase in the Bax/Bcl-2 ratio (Figure 6i, *p* < 0.01). Notably, overexpression of RET mitigated the increase in apoptosis induced by METH (Figure 6h–k, *p* < 0.01 or *p* < 0.05).

Furthermore, METH exposure caused an accumulation of GFRα1 on the cell membrane (Figure 6e, *p* < 0.01) and impaired GDNF-GFRα1 endocytosis (Figure 6f, *p* < 0.05), without affecting GFRα1 expression (Figure 6d, *p* > 0.05). RET overexpression attenuated METH-induced GFRα1 membrane accumulation (Figure 6e, *p* < 0.05) and partially restored GDNF-GFRα1 endocytosis (Figure 6f, *p* < 0.05). These results suggest that RET-regulated GDNF-GFRα1 endocytosis plays a critical role in METH-induced neurotoxicity.

## 3. Discussion

Analysis of human post-mortem hippocampal tissue revealed that METH abuse led to pronounced neuronal apoptosis, contributing to neurotoxicity. However, METH exposure also induced a compensatory increase in GDNF expression, suggesting an endogenous neuroprotective response. In vitro experiments further demonstrated that METH exposure resulted in a progressive decline in cell activity and the loss of normal cellular morphology in both co-culture systems in a dose- and time-dependent manner. Concurrently, GDNF expression and release were significantly upregulated. Notably, METH disrupted GDNF-GFRα1 endocytosis in HT-22 cells, while overexpression of RET mitigated METH-induced cytotoxicity and restored GDNF-GFRα1 endocytosis. Mouse experiments corroborated these results, showing that METH exposure increased apoptosis, elevated GDNF expression, and down-regulated RET expression in hippocampal neurons. These findings suggest that RET-mediated GDNF-GFRα1 endocytosis is a critical regulatory mechanism in METH-induced neurotoxicity. Impairment of this pathway may compromise the neuroprotective function of GDNF, exacerbating neuronal damage. But it is not clear that compromised RET-mediated GDNF-GFRα1 endocytosis is the mechanism of METH-induced neurotoxicity. 

METH-induced neurotoxicity is a multifaceted process involving oxidative stress, apoptosis, mitochondrial dysfunction, endoplasmic reticulum dysfunction, and aberrant activation of glial cells. One primary mechanism of METH-induced neuronal damage is oxidative stress, driven by excessive dopamine (DA) accumulation. METH competitively binds to dopamine transporter (DAT) and vesicular monoamine transporter 2 (VMAT-2), leading to DA accumulation inside and outside neurons. This excess DA undergoes auto-oxidation, generating high levels of reactive oxygen species (ROS) and triggering severe oxidative stress [3,4,26,27]. Beyond oxidative damage, METH-induced DA accumulation contributes to mitochondrial dysfunction, with DA oxidation byproducts disrupting mitochondrial homeostasis. Studies have shown that METH exposure leads to mitochondrial swelling and electron transport chain disruption [28,29]. Additionally, glutamate receptor activation and peroxynitrite generation exacerbate mitochondrial impairment, ultimately initiating apoptosis [30]. METH exposure up-regulates pro-apoptotic proteins (Bax) while down-regulating anti-apoptotic proteins (Bcl-2) in rodent brains, further promoting neuronal loss [31,32,33,34,35]. Bcl-2 anti-apoptotic protein is mainly located on the outer membrane of mitochondria and prevents apoptosis by inhibiting the release of apoptotic factors (such as cytochrome c) from mitochondria. Bax pro-apoptotic protein, mainly exists in the cytoplasm and is translocated to mitochondria under apoptotic signal stimulation, promoting membrane pore formation and cytochrome c release into the cytoplasm. The translocation of Bax from the cytoplasm to mitochondria is a crucial step in initiating apoptosis. At this point, the ratio of Bax/Bcl-2 in mitochondria increases, indicating an increase in cell apoptosis. Bax decreases in the cytoplasm, and the Bax/Bcl-2 ratio decreases. However, our study did not separately detect the expression of Bax and Bcl-2 in mitochondria and cytoplasm, which is a limitation of our research [36,37,38,39]. Cytochrome c released into the cytoplasm interacts with apoptotic peptidase-activating factor 1 and caspase-9 to form the apoptosome, initiating a caspase cascade. This cascade sequentially activates caspase-9 and downstream effector caspases, such as caspase-3, culminating in neuronal apoptosis [40,41]. In this study, METH-induced apoptosis was confirmed in both human and mouse hippocampal tissues, as well as in in vitro models. Notably, RET overexpression in HT-22 cells significantly reduced METH-induced apoptosis in vitro, suggesting a potential protective role. However, the precise molecular mechanisms underlying RET-mediated neuroprotection remain unclear and warrant further investigation.

Several studies have demonstrated that GDNF family ligands, such as neurturin and artemin, members of the GDNF family, effectively counteract METH-induced dopaminergic neurotoxicity. Other neurotrophic factors, including brain-derived neurotrophic factor (BDNF), NT-3, acidic fibroblast growth factor, basic fibroblast growth factor, ciliary neurotrophic factor, transforming Growth Factor-α (TGF-α), heregulin β1 (HRG-β1), and amphiregulin (AR), have also been shown to exert significant protective effects [42]. Consistent with these findings, the present study demonstrated that METH exposure led to a dose-dependent increase in GDNF expression and release in vitro, as observed in human post-mortem samples and in vivo mouse models. Despite this compensatory up-regulation of GDNF, previous studies have shown that in Parkinson’s disease models induced by 6-OHDA and MPTP, RET expression is significantly reduced, impairing GDNF-mediated neuroprotection [23,24,43].Consistent with this, our findings revealed that METH exposure disrupted GDNF-GFRa1-RET endocytosis in HT-22 cells, leading to impaired intracellular signaling. While total GFRα1 protein expression remained unchanged, RET expression was significantly down-regulated, further compromising the protective effects of GDNF. Overexpression of RET restored GDNF-GFRα1-RET endocytosis, suggesting that METH-induced impairment of this pathway is mediated through RET down-regulation. There is also evidence to suggest that in the absence of RET, GDNF-GFRα1 also undergoes cellular signaling, and neuronal adhesion molecules (NCAM) and synaptophysin III can mediate the endocytosis of GDNF-GFRα1, thereby exerting their protect effects [44]. At the same time, there is evidence to suggest that the number of NCAM-positive cells in the hippocampus of one-day-old juvenile rats exposed to METH during pregnancy is significantly reduced, and the intensity of molecular expression is also significantly reduced. In 22-day-old rats, the number of NCAM-positive cells and the intensity of expression of these molecules in the hippocampus are also significantly reduced [45]. Moreover, METH can induce a decrease in synaptophysin proteins in different brain regions [46]. This suggests that METH may lead to endocytic disorders of GDNF-GFRα1 through various mechanisms. However, GDNF-GFRα1 RET remains a classic endocytic pathway. Taken together, this study provides new insights into METH-induced neurotoxicity by demonstrating that disrupted GDNF-GFRα1 endocytosis contributes to neuronal damage. These findings highlight the role of RET-regulated receptor trafficking in modulating METH-induced neurodegeneration, offering a foundation for future investigations into therapeutic strategies aimed at mitigating METH-associated neurotoxicity and withdrawal symptoms.

There is sufficient evidence to suggest that repeated exposure to METH can lead to neurotoxicity and induce neurodegenerative manifestations, including Parkinson’s disease (PD) [6] and Alzheimer’s disease (AD) [7]. METH administration can lead to the death of midbrain dopaminergic neurons and the formation of characteristic inclusion bodies formed by the aggregation of alpha synuclein, which is a feature of Parkinson’s disease [47,48,49]. METH abuse has been shown to significantly upregulate amyloid precursor protein (APP) and phosphorylated tau protein [50], as well as reduce hippocampal volume and neuronal degeneration, leading to memory loss and impaired learning ability [51]. This is consistent with the characteristics and clinical manifestations of AD. In our study, concentration- and time-dependent degeneration was observed in HT-22 cells after METH exposure. Provide more evidence for METH induced neurodegenerative changes.

Currently, RET is considered an attractive drug target for the treatment of neurodegenerative diseases (including AD and PD) and amyotrophic lateral sclerosis [52,53]. In addition, it is worth noting that the interaction between clathrin-coated pits (CCP) and adaptor protein complex 2 (AP2) promotes the internalization of RET receptors through clathrin-mediated endocytosis [17]. Some studies have also found that RET can be detected in both early and late endosomes of neuronal cells, indicating that RET can also be internalized through the endosomal pathway [54]. Subsequently, studies have shown that METH hinders membrane fusion and differentiation during endosome maturation, which may also be one of the ways in which METH induces endocytosis disorders in GDNF-GFRα1-RET [55]. A study found that acute METH exposure increased the expression of GDNF in the striatum of mice, while chronic METH exposure decreased the expression of GDNF receptors GFRα1 and RET in the striatum [56]. In our experimental results, acute METH exposure induced an increase in GDNF expression and a decrease in RET expression in the hippocampus of mice, indicating that differences in RET expression may be related to different functions of mouse brain regions. It is worth noting that GDNF-GFRα1-RET can promote the interaction between the key signaling pathways PI3K/AKT and ERK1/2. The dysregulation of these pathways is associated with the occurrence of neurodegenerative diseases. Research has shown that GDNF activation of RET receptors can enhance AKT signaling and promote cell survival by inhibiting the apoptotic pathway. Our research indicates that exposure to METH leads to increased cell apoptosis, decreased cell survival rate, and impaired endocytosis of GDNF-GFRα1-RET, which may be the reason why GDNF does not exert an inhibitory effect on apoptosis. However, this study did not validate its downstream key signaling pathways, which may be a limitation of this study. In addition, the inhibitory effect on Glycogen Synthase Kinase 3β (GSK3 β, inhibiting its activity can promote cell survival) has been shown to potentially improve the pathology associated with tau, while small-molecule RET agonists may enhance therapeutic efficacy [57]. This suggests that emerging methods such as gene therapy and small-molecule RET agonists may provide new avenues for treating METH and neurodegenerative diseases.

This study has some limitations and is worth further exploration. Although we observed the effects of METH on GDNF and its signaling pathways, we did not delve into the interactions of its downstream signaling pathways, such as the PI3K/AKT and ERK1/2 signaling pathways. This may be a limitation of this study. Finally, this study did not fully consider clinically relevant METH exposure patterns to better assess their potential impact on the human nervous system.

## 4. Materials and Methods

### 4.1. Drugs and Post-Mortem Brain Samples

Methamphetamine hydrochloride was procured from the Yunnan Institute for Drug Abuse (Kunming, China) and dissolved in phosphate-buffered saline (PBS; pH 7.2; Servicebio, G4202, Wuhan, China) before use. Post-mortem brain samples from METH users (*n* = 5) and control individuals (*n* = 5) were obtained from the Forensic Appraisal Center of Kunming Medical University. Inclusion criteria for METH users included: (1) male sex; (2) age between 18 and 50 years; (3) detection of METH in blood at the time of death; and (4) a documented history of METH use. Control individuals died from cardiovascular disease and had no recorded history of METH use.

Brain tissue collection was performed with the signed informed consent of the deceased’s family, adhering to the Code of Ethics of the World Medical Association. Five cases of METH abuse deaths and five cases of exclusion of central nervous system diseases and injuries deaths, with 5 samples per group frozen for protein extraction and 5 samples fixed with formaldehyde for sectioning. All procedures were approved by the Medical Ethics Committee of Kunming Medical University (permit number: KMMU2022MEC151). For protein analysis, the hippocampus was dissected from post-mortem brains and stored at −80 °C. For histological analysis, hippocampal tissue was fixed in 4% paraformaldehyde (Servicebio, G1101, Wuhan, China) and dehydrated in increasing concentrations (10%, 20%, and 30%) of sucrose solution (Beyotime, ST1670, Shanghai, China). Tissue sections (5 μm) were obtained using a freezing microtome (RWD Life Science, FS800A, Shenzhen, China).

### 4.2. Terminal Deoxynucleotidyl Transferase dUTP Nick-End Labeling (TUNEL) Staining

Apoptotic cells were detected using a TUNEL staining kit (Sigma-Aldrich, 12156792910 or 11684817910, St. Louis, MO, USA) following the manufacturer’s protocols. Tissue sections were fixed in 4% paraformaldehyde at room temperature for 20 min, followed by three washes with PBS for 5 min each. Permeabilization was performed by incubating the sections in PBS containing 2‰ Triton X-100 (Solarbio Life Sciences, T8200, Beijing, China) and 1‰ sodium citrate (Solarbio Life Sciences, YS175276, Beijing, China) at room temperature for 10 min, followed by three additional PBS washes. The sections were then incubated with the TUNEL reaction mixture in a humidified chamber at 37 °C for 60 min and subsequently washed three times with PBS. Nuclear counterstaining was performed using 4’, 6-diaminidine 2-phenylindole (DAPI, Solarbio Life Sciences, S2110, Beijing, China) for 20 min. Images were acquired using a fluorescence microscope (Frankfurt, Germany), and TUNEL-positive cells were quantified using ImageJ (v1.46r).

### 4.3. Immunohistochemical Staining

Tissue sections were washed with PBS for 3 min before antigen retrieval was performed using a 5× sodium citrate antigen repair solution (Solarbio Life Sciences, C1032, Beijing, China). The sections were heated in a pressure cooker until steam was released for 4 min, after which heating was discontinued, and the samples were allowed to cool naturally. Sections were subsequently washed in PBS for 3 min and treated with a peroxidase blocker (Zhongshan Jinqiao, PV9000, Beijing, China) for 10 min at room temperature, followed by another 3-minute PBS wash. Permeabilization was carried out using 0.2% TritonX-100 for 20 min, and non-specific binding was blocked with 10% or 20% goat serum (Solarbio Life Sciences, SL038, Beijing, China) at room temperature for 1 h.

For immunolabeling, sections were incubated overnight at 4 °C with primary antibodies against GDNF (Abcam, ab176564, 1:200, Wuhan, China) or RET (Abcam, ab134100, 1:200, Wuhan, China) in PBS, followed by three PBS washes. A reaction-enhancing solution (Zhongshan Jinqiao, PV9000, Beijing, China) was applied for 40 min before another three washes with PBS. Sections were then incubated with an enhanced enzyme-labeled sheep anti-mouse/rabbit IgG polymer (Zhongshan Jinqiao, PV-9000, Beijing, China) at room temperature for 40 min. Detection was performed using a diaminobenzidine (DAB) color development solution (Zhongshan Jinqiao, ZLI-9018, Beijing, China), with reaction times optimized for each antibody. The slices were successively placed in 50% anhydrous ethanol (20 s), 70% anhydrous ethanol (20 s), 80% anhminydrous ethanol (20 s), 90% anhydrous ethanol (20 s), 95% anhydrous ethanol (20 s), anhydrous ethanol (20 s), and xylene (20 s) for gradient dehydration and transparency. Finally, the samples were sealed using neutral gum mixed with xylene and stored appropriately. Images were then scanned using a KF-PRO-005-EX scanner (Ningbo, China).

### 4.4. Western Blotting

Brain tissues or cultured cells were cleaved on ice for 30 min using RIPA lysis buffer (Epizyme Biotech, PC101, Shanghai, China). After centrifugation at 12,000 rpm for 15 min at 4 °C, the protein concentration in the supernatant was detected using a bicinchoninic acid (BCA) protein concentration assay kit (Epizyme Biotech, ZJ102, Shanghai, China). For biotinylated samples, protein concentrations were determined directly. Proteins were isolated by polyacrylamide gel electrophoresis (PAGE), using gels of varying concentrations, depending on the molecular weight of the target protein (Epizyme Biotech, PG213, Shanghai, China), and subsequently transferred onto polyvinylidene fluoride (PVDF) membranes (Sigma-Aldrich, IPVH00010, MO, USA). The membranes were blocked with 5% skim milk prepared in Tris-buffered saline (Epizyme Biotech, PS112, Shanghai, China) for 2 h at room temperature. Primary antibody incubation was performed overnight at 4 °C using the following antibodies: GDNF (Abcam, ab176564, 1:1000, Wuhan, China), RET (Abcam, ab134100, 1:1000, Wuhan, China), Gfra1 (Abcam, ab186855, 1:1000, Wuhan, China), BCL-1 (Proteintech, 60178-1-IG, 1:1000, Wuhan, China), Bax (Proteintech, 60267-1-IG, 1:2000, Wuhan, China), Caspase-3 (Proteintech, 19677-1-AP, 1:700, Wuhan, China), lysed Caspase-3 (Proteintech, 25128-1-AP, 1:700, Wuhan, China), and β-actin (Proteintech, 66009-1, 1:5000, Wuhan, China). The membranes were then rinsed three times in Tris-buffered saline containing 2‰ Tween-20 (TBST) for 15 min per wash. The membranes were then incubated with either a secondary antibody (Signalway Antibody, #L3012, 1:5000, College Park, MD, USA) or horseradish peroxidase (HRP)-conjugated anti-mouse antibody (#L3032, 1:5000, MD, USA) for 2 h at room temperature, followed by three additional TBST washes (15 min each). Immunoreactive proteins were detected using an ECL chemiluminescent substrate kit (Biosharp Life Sciences, BL520B, Hefei, China). Images were captured using an imaging system (Bio-Rad Laboratories, ChemiDoc MP, Hercules, CA, USA) and analyzed using ImageJ (v1.46r).

### 4.5. Cell Culture and Treatments

HT-22 cells were purchased from iCell Bioscience (CVCL_0321, Shanghai, China), BV2 cells were obtained from the Cell Resource Center, Shanghai Institute of Biological Sciences, Chinese Academy of Sciences (Shanghai, China), and MA cells were purchased from Kunming Jiejing Meiya Biotechnology Co., Ltd., Kunming, China. These three cell types were co-cultured at a ratio of 5:2:1 in Dulbecco’s Modified Eagle Medium (DMEM) (Servicebio, G4515, Wuhan, China) containing 10% fetal bovine serum (VivaCell, C04001, Shanghai, China) and 1% penicillin/streptomycin (Solarbio Life Sciences, P1400, Beijing, China). Cells were maintained in a humidified incubator (90%) at 37 °C under 5% CO_2_.

For immunofluorescence assays, a Transwell co-culture system was used, while direct contact co-culture was used for all other cell-based experiments. To determine the optimal concentration and duration of METH exposure for inducing cellular damage, cells were treated with a concentration gradient of METH (0.25, 0.5, 1, 2, or 4 mM) for 24 h. The optimal concentration was then used to assess exposure duration, with time points of 3, 6, 12, 24, and 48 h.

### 4.6. Immunofluorescence Staining

HT-22 and MA cells, as well as tissue sections, were fixed in 4% paraformaldehyde for 30 min, followed by three gentle washes with PBS. Permeabilization was performed using 2‰ Triton X-100 for 10 min, after which samples were sealed with 10% or 20% goat serum at room temperature for 1 h. For immunolabeling, samples were incubated overnight at 4 °C with primary antibodies targeting TUBB3 (Proteintech, 66375-1-Ig, 1:200, Wuhan, China), GDNF (Abcam, ab176564, 1:200, Wuhan, China), or RET (Abcam, ab134100, 1:200, Wuhan, China) in PBS. After three PBS washes, sections were incubated for 1.5 h at 37 °C with either Alexa Fluor 594-conjugated goat anti-rabbit secondary antibody (Invitrogen, A-11012, 1:200, Carlsbad, CA, USA) or Alexa Fluor 488-conjugated goat anti-mouse secondary antibody (Invitrogen, A-11001, 1:200, CA, USA) in PBS. Samples were then washed three times with PBS, and nuclei were labeled with a DAPI-containing sealer for 20 min. Images were acquired using a fluorescence microscope, and cell morphology parameters (width and length), as well as fluorescence intensity, were quantified using ImageJ (v1.46r).

### 4.7. Enzyme-Linked Immunosorbent Assay (ELISA)

The supernatant from co-cultured cells exposed to METH was collected, and GDNF levels were detected using an ELISA kit (Enzyme-linked Biotechnology, Shanghai, China), following the manufacturer’s instructions. Immunolabeling of GDNF protein was performed as per the kit protocols. Absorbance was measured at 450 nm using a microplate reader, and GDNF concentration was calculated based on the standard curve and formula.

### 4.8. Cell Viability Assay

Cell viability was assessed using a CCK-8 cytotoxicity assay kit (Absin, abs50003, Shanghai, China). HT-22, MA, and BV2 cells were seeded in 96-well plates (NEST Biotechnology, 713011, Wuxi, China) at a ratio of 5:2:1. Following cell adhesion, METH treatment was applied, and the cells were incubated with 10% CCK-8 reagent at 37 °C for 60 min. Absorbance was measured at 450 nm using a microplate reader, and cell activity was calculated according to the manufacturer’s guidelines.

### 4.9. Extraction of Cell Membrane Proteins

Cell membrane proteins were isolated using a membrane protein extraction kit (Beyotime, P0033, Shanghai, China) following the manufacturer’s protocols. Briefly, HT-22 cell homogenates were prepared by adding reagent A and manually homogenizing 20–30 times using a pre-cooled glass homogenizer. The homogenates were incubated on ice for 30 min and centrifuged at 700× *g* for 10 min at 4 °C. The supernatant was further centrifuged at 14,000× *g* for 30 min at 4 °C, with the resulting pellet precipitated with reagent B and placed on ice for 30 min, followed by Western blot.

### 4.10. Biotinylation Assay

HT-22 cells were incubated in Hank’s balanced salt solution (HBSS, Sigma-Aldrich, H6648, MO, USA) containing 1 mg/mL of sulfo-NHS-SS-biotin (MedChemExpress, HY-111496, Shanghai, China) at 4 °C for 30 min to biotinylate membrane proteins. Cells were then washed three times with HBSS, and biotinylation was terminated with 20 mM glycine buffer at 4 °C for 15 min. After two washes with DMEM, cells were treated with GDNF protein (HY-P7359, MedChem Express, Monmouth Junction, NJ, USA) for 2 h, followed by METH exposure for 24 h. Subsequently, the HT-22 cells were lysed in RIPA buffer at 4 °C for 30 min and centrifuged at 12,000 rpm and 4 °C for 15 min. The supernatant was incubated with NeutrAvidin^®^ UltraLink^®^ resin (Thermo Fisher Scientific, 53150, Waltham, MA, USA) in a polypropylene column (Thermo Fisher Scientific, 29922, MA, USA) to isolate biotinylated proteins. Elution was performed using 8 M guanidine hydrochloride buffer (Thermo Fisher Scientific, 24115, MA, USA) at pH 1.5, yielding purified biotinylated proteins, which were subsequently analyzed by Western blot.

### 4.11. Construction of Cell Lines Overexpressing RET

Lentiviral plasmids PCSLENT-CMV-MCS-3Xflag-PGK-Puro-Wpre3 (titer: 7.12 × 10^8^ TU/mL) and PCSLENT-CMV-RET-3xflag-PGk-Puro-Wpre3 (titer: 2.32 × 10^8^ TU/mL) were prepared and packaged by OBiO Technology (HY2208452, Shanghai, China). HT-22 cells in the positive experimental group were infected with lentivirus (LV) for 12 h at multiplicities of infection (MOIs) of 20, 40, or 80, while control cells were infected with the negative control LV (MOI 80) under identical conditions. Following infection, the culture medium was replaced, and transfected cells were purified using puromycin (2 μg/mL for 4 days) 48 h after LV infection. Finally, RET overexpression was confirmed via Western blot analysis.

### 4.12. Mouse Hippocampal Tissue Collection and Brain Sectioning

Male C57BL/6 mice (8 weeks old, *n* = 20) were purchased from the Laboratory Animal Center of Kunming Medical University, China. The mice were housed under controlled conditions (humidity: 50% ± 10%; temperature: 22 ± 1 °C) with free access to food and water under a 12 h reverse light/dark cycle. All animal procedures were carried out in accordance with the Guidelines for the Care and Use of Laboratory Animals of the National Research Council and approved by the Animal Care and Use Committee of Kunming Medical University (License number: KMMU20220735). The authors also ensured compliance with the ARRIVE guidelines. After a one-week acclimation period, the mice were assigned to either the normal saline group or the METH treatment group. Mice in the METH group received intraperitoneal injections of METH (10 mg/kg) every 3 h for a total of four doses, establishing a binge-like METH administration model to induce acute neurotoxicity [43,58,59,60,61].Control mice received equivalent volumes of saline. Twenty-four hours post-treatment, mice were anesthetized with isoflurane, then injected with physiological saline into the heart to fully empty the entire body of blood, and hippocampal tissue was collected. For protein analysis, hippocampal samples were stored at −80 °C, while tissue sections were fixed in 4% paraformaldehyde and dehydrated in increasing concentrations of sucrose. Coronal hippocampal sections (10 μm) were prepared using a cryotome.

### 4.13. Statistical Analysis

All statistical analyses were performed using SPSS v26.0, with data presented as mean ± standard deviation (SD). Independent samples *t*-tests were used for comparisons between two groups. One-way analysis of variance (ANOVA) was used to analyze data from screening experiments on METH concentrations and time points, as well as RET transcription detection. Two-way ANOVA was used to analyze data from other cell-based experiments. Post hoc analyses were conducted using Tukey’s honest significant difference (HSD) test. Normality and homogeneity of variance were confirmed prior to analysis. The level of significance was set to *p* < 0.05.

## 5. Conclusions

This study draws two conclusions: one is that methamphetamine induces neurotoxicity and GDNF-GFRα1 endocytosis disorder. Secondly, the endocytosis disorder of GDNF-GFR α 1 regulated by transmembrane receptor tyrosine kinase (RET), promotes methamphetamine induced neurotoxicity.

## Figures and Tables

**Figure 1 ijms-26-09522-f001:**
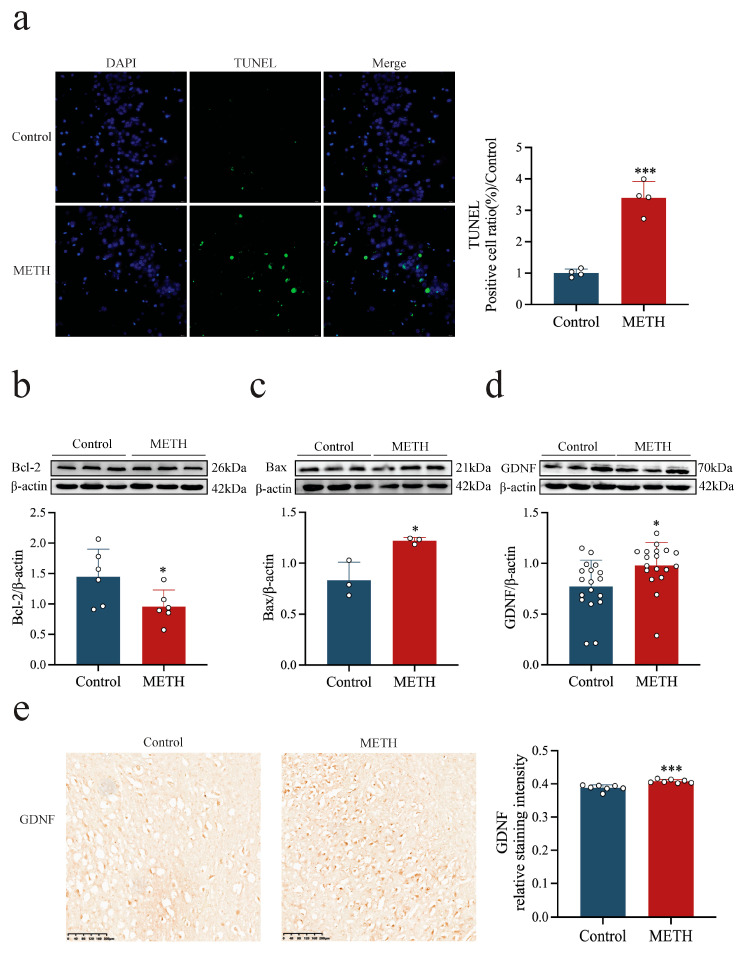
METH induces apoptosis and GDNF expression in the human hippocampus. (**a**) Apoptosis levels in the DG region of the human hippocampus. scale bar = 20 μm. (**b**–**e**) Expression levels of apoptosis-related proteins and GDNF in the human hippocampus. * *p* < 0.05, *** *p <* 0.001 compared to control group. Statistical significance was determined using an independent samples *t*-test.

**Figure 2 ijms-26-09522-f002:**
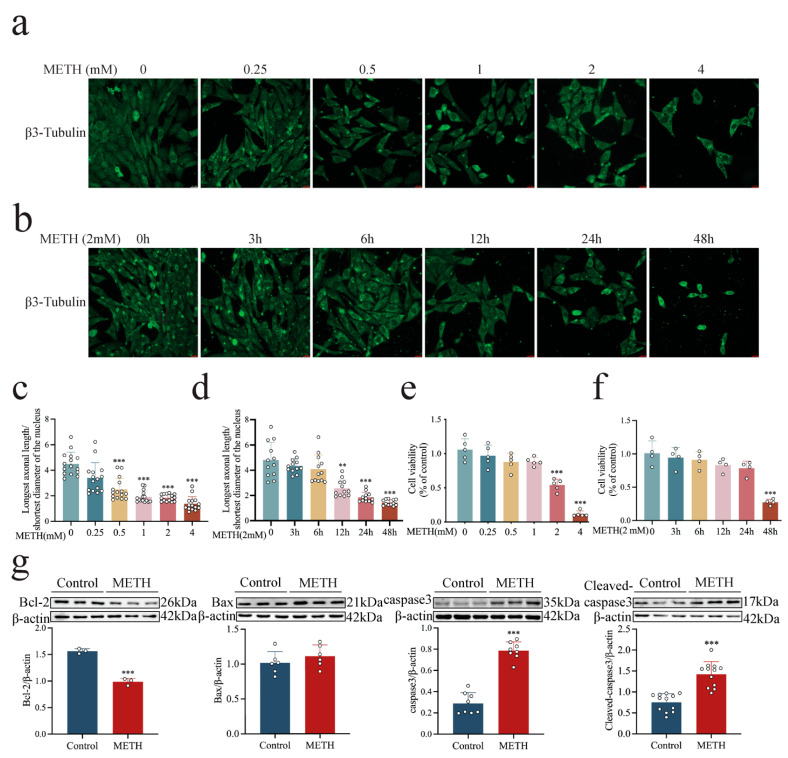
METH induces neurotoxicity in HT-22 cells. (**a**) Representative images showing HT-22 cell morphology following 24 h exposure to increasing concentrations of METH. scale bar = 20 μm. (**b**) Representative images showing HT-22 cell morphology following exposure to METH (2 mM) for increasing durations. scale bar = 20 μm. (**c**,**d**) The ratio of the long axis to the short axis of HT-22 cells gradually decreases as the concentration and duration of methamphetamine action gradually increase. (**e**) Cell viability of co-cultured cells following 24 h exposure to increasing concentrations of METH. (**f**) Cell viability of co-cultured cells following exposure to METH (2 mM) for increasing durations. (**g**) Expression levels of apoptosis-related proteins in co-cultured cells following 24 h exposure to 2 mM METH. ** *p* < 0.01, *** *p* < 0.001 compared to control group. Statistical significance was determined using ANOVA, followed by Tukey’s HSD post hoc test or an independent samples *t*-test.

**Figure 3 ijms-26-09522-f003:**
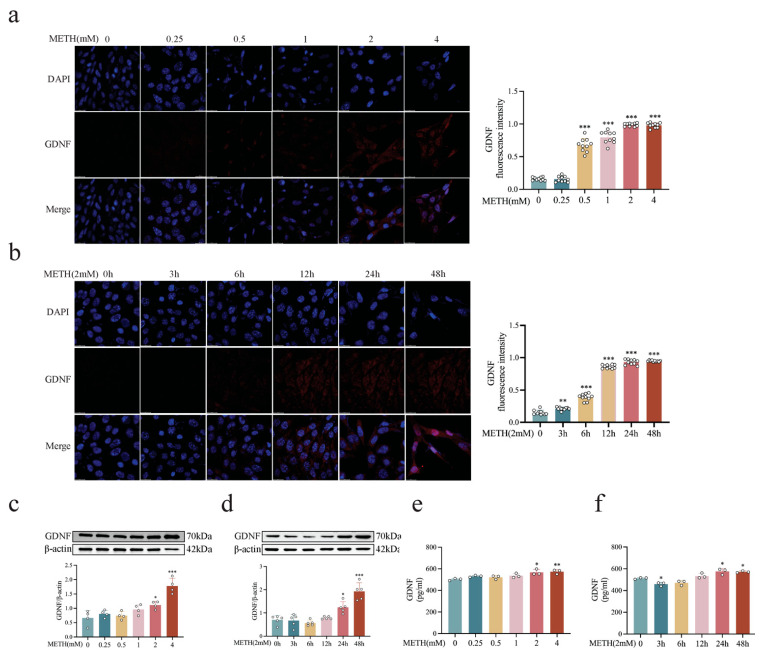
METH induces GDNF expression and release in co-cultured cells. (**a**,**c**) GDNF expression levels in co-cultured cells following 24 h exposure to increasing concentrations of METH. scale bar = 20 μm. (**b**,**d**) GDNF expression levels in co-cultured cells following exposure to METH (2 mM) for increasing durations. scale bar = 20 μm. (**e**) GDNF release in co-cultured cells following 24 h exposure to increasing concentrations of METH. (**f**) GDNF release in co-cultured cells following exposure to METH (2 mM) for increasing durations. * *p* < 0.05, ** *p* < 0.01, *** *p* < 0.001 compared to control group. Statistical significance was determined using ANOVA, followed by Tukey’s HSD post hoc test.

**Figure 4 ijms-26-09522-f004:**
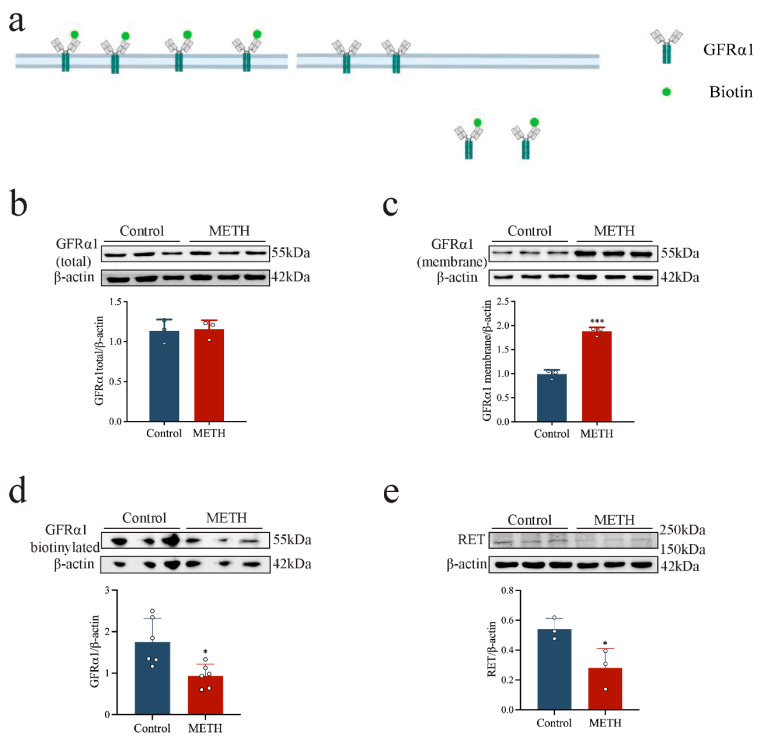
METH impairs GFRα1 endocytosis. (**a**) Biotinylation assay demonstrating GFRα1 endocytosis. (**b**–**d**) Expression levels of total GFRα1, membrane GFRα1, and biotinylated GFRα1 in HT-22 cells following 24 h exposure to 2 mM METH. (**e**) RET expression levels in HT-22 cells after 24 h incubation with 2 mM METH. * *p* < 0.05, *** *p* < 0.001 compared to control group (independent sample *t*-test).

**Figure 5 ijms-26-09522-f005:**
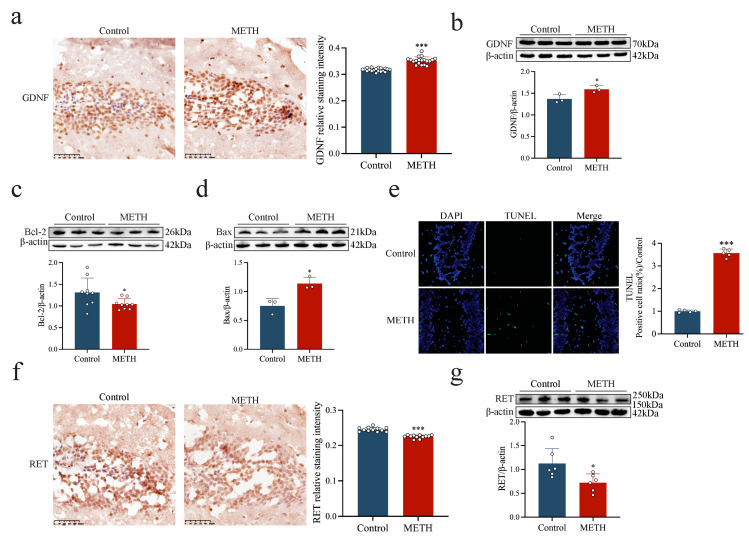
METH induces neurotoxicity and increases GDNF and RET expression in mouse hippocampal tissue. (**a**,**b**) Expression of GDNF in hippocampal tissue 24 h after METH administration. (**c**–**e**) Expression of apoptosis-related proteins in hippocampal tissue 24 h after METH administration. scale bar = 20μm. (**f**,**g**) RET expression in hippocampal tissue 24 h after METH administration. * *p* < 0.05, *** *p* < 0.001 compared to control group. Statistical significance was determined using an independent samples *t*-test.

**Figure 6 ijms-26-09522-f006:**
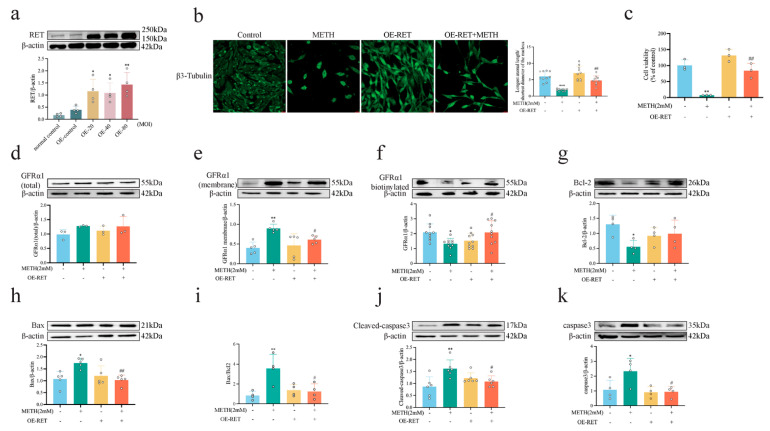
RET overexpression mitigates METH-induced neurotoxicity and GFRα1 endocytosis in HT-22 cells. (**a**) RET expression levels following LV transfection. (**b**) Representative images of HT-22 cell morphology. scale bar = 20 μm. (**c**) Cell viability in co-cultured cells. (**d**–**f**) Expression levels of total GFRα1, membrane GFRα1, and biotinylated GFRα1 proteins. (**g**–**k**) Expression levels of apoptosis-related proteins. * *p* < 0.05, ** *p* < 0.01, *** *p* < 0.001 compared to control group (two-way ANOVA); ^#^ *p* < 0.05 or ^##^ *p* < 0.01 compared to METH group (two-way ANOVA).

## Data Availability

The datasets used and/or analyzed during the current study are available from the corresponding author on reasonable request.

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
