# Peer review of "Role of RET-Regulated GDNF-GFRα1 Endocytosis in Methamphetamine-Induced Neurotoxicity"

_ijms, 2025, doi:10.3390/ijms26199522_

Round 1

Reviewer 1 Report

Comments and Suggestions for Authors

The manuscript entitled “Role of RET-regulated GDNF-GFRα1 endocytosis in methamphetamine-induced neurotoxicity” addresses an important and timely topic in neurobiology. Methamphetamine is a psychostimulant with severe neurotoxic effects that contribute to neuropsychiatric disorders, making studies of its mechanisms highly relevant, however, the authors should address several issues to improve the overall quality, clarity, and transparency of the manuscript.

The abstract is generally clear and informative, but it could be written in a more engaging and impactful way to better reflect the scope of the study. At present, certain formulations are repetitive (e.g., the phrase “this study” appears more than once), which reduces readability. A more concise restructuring, with stronger emphasis on the novelty of findings and their broader implications, would make the abstract more attractive. The Introduction is well structured and provides the necessary background, with clearly defined aims of the study. 

With regard to the Results section, several improvements are needed. It would be beneficial to include individual data points within the bar graphs (e.g., scatter dots overlaid on the bars), as this would enhance transparency and allow readers to assess data variability. In Figure 1, TUNEL staining and the immunohistochemistry for GDNF should be presented separately, as the current combined format is overcrowded and diminishes clarity. The figure legend should be expanded to clearly describe each panel (a–e). Additionally, there is a labeling error in Figure 1: the panel designation “b” is duplicated while “d” is missing, which should be corrected.

The Materials and Methods section also requires clarification. The number of human post-mortem samples used is not sufficiently specified. If a total of five human samples were included, it should be clearly stated how many were analyzed by blotting and how many by IHC. This is crucial for assessing the strength and relevance of the human data presented and directly impacts the validity of the study’s conclusions.

The Discussion, although comprehensive, comes across as somewhat mechanical and would benefit from improved style and narrative flow. Strengthening the interpretation and better highlighting the broader significance of the findings would allow the main message of the study to come across more strongly. Furthermore, certain methodological details (e.g., descriptions of membrane protein fractionation and biotin-based receptor labeling) should be placed in the Materials and Methods section rather than in the Discussion, where they distract from interpretation. Importantly, the limitations of the study should be addressed more explicitly. In particular, the analysis of Bax and Bcl2 is insufficiently elaborated. It would be essential to compare not only their relative expression levels but also their distribution in mitochondrial fraction versus cytosolic fraction, since this ratio is highly relevant for determining apoptotic processes.

In summary, the study addresses a significant research question and provides valuable mechanistic insights into methamphetamine-induced neurotoxicity through the disruption of GDNF–GFRα1–RET signaling. The experimental design is comprehensive, combining human data with in vivo and in vitro approaches, which strengthens the translational impact of the findings. However, several aspects require revision, particularly in the presentation of figures, methodological clarity, discussion style, and addressing of study limitations. If these points are adequately revised, the manuscript will be substantially improved in terms of quality, transparency, and overall scientific impact.

Author Response

Please see the attachment. And the revised manuscript has been resubmitted

Reviewer 2 Report

Comments and Suggestions for Authors

The authors studied the involvement of GDNF-GFRα1-RET complex impairment in the mechanism of methamphetamine-induced neurotoxicity. The topic is interesting and relevant as methamphetamine abuse is a global public health problem. Unrevealing the intimate mechanisms of methamphetamine-induced neurotoxicity is the scientific base for the development of pharmacological strategies for its prevention. The neurotoxic effects of methamphetamine have been investigated on human post-mortem hippocampal samples, co-culture system of mouse microglial, astroglial and hippocampal neuronal cell lines, and in a mouse model of acute methamphetamine toxicity. The manuscript is well-organized. The background is well written and justifies the aim of the study, which is explicitly defined.The methods are described in sufficient detail to ensure the reproducibility of the results. I have the following minor comments concerning the presentation of the results and the discussion section:

  • Line 88 – Fig. 1b should be corrected to 1c (expression of Bax) and conversely in line 89 fig. 1c should be corrected to 1b (expression of Bcl-2). In fig. 1 the image with the expression of GDNF should be “d”, not “b” (typographical error).
  • Line 106 – The concentration of 0.25 does not demonstrate statistical significance; therefore, this value should be omitted.
  • Lines 107–108 – Based on the results presented in Figure 2, the sentence should read: “METH reduced cell viability in a concentration-dependent manner, with statistical significance reached at concentrations of 2 and 4 mM ( 2c, P < 0.001)”
  • Line 110 – “(Fig. 2b, P < 0.001)” should be corrected to ( 2b, P<0,01 or P < 0.001).
  • Line 111 – should be clarified that there is significant difference only at 48 h.
  • Line 125 “(Fig. 3a,b, P < 0.001)” should be corrected to ( 3a,b, P<0,01 or P < 0.001).
  • Line 150 “(Fig. 4a, d, P < 0.001)” should be corrected to ( 4a,d, P<0,05) and in Fig. 4d biotinylated should be added below GFRα1.
  • The caption of fig. 5 “METH induces neurotoxicity and increases GDNF and RET expression in mouse hippocampal tissue” should be corrected to “…increases GDNF and decreases RET expression…”.
  • Lines 189-190 “…while RET overexpression increased the long-to-short axis ratio ( 6b, P < 0.01) 189 and improved cell activity (Fig. 6c, P < 0.01)”. It should be clarified that this effect is observed in the presence of METH and is based on comparisons with METH-treated samples rather than the control.
  • Line 192 “( 191 6h, j, k, P < 0.01) should be corrected to (Fig. 191 6h, j, k, P < 0.05 or < 0.01).
  • Line 194 ( 6g, h, i, j, k, P < 0.01 or P < 0.05). "g" should be omitted, as no statistical significance is demonstrated based on the data presented in Figure 6.
  • Line 197 ( 6d, P < 0.05) should be corrected to (Fig. 6d, P > 0.05) as there is no significant difference.

Discussion section

Line 220 “These findings suggest that RET-mediated GDNF-GFRα1 endocytosis is a critical regulatory mechanism in METH-induced neurotoxicity” – it is not clear that compromised RET-mediated GDNF-GFRα1 endocytosis is the mechanism of METH-induced neurotoxicity. This should be emphasized.

The second paragraph of the Discussion section appears more suitable for the Introduction, as it justifies the selection of methods used to achieve the study’s aim and outlines their advantages. I recommend including a Conclusion section following the Discussion.
Additionally, the sections on Author Contributions, Funding, and related information seem to be misplaced, likely due to a formatting error, and should be repositioned accordingly.

Author Response

Please see the attachment. The manuscript has been revised and resubmitted
